environmental chemistry/environmental science/environmental engineering

RoHS2, phthalate ester, bromodiphenyl ether, thermal desorption mass spectroscopy, atmospheric pressure chemical ionization

**Author for correspondence:**
Masataka Ohgaki
e-mail: masataka.ogaki.fu@hitachi-hightech.com

This article has been edited by the Royal Society of Chemistry, including the commissioning, peer review process and editorial aspects up to the point of acceptance.

# Screening analysis of RoHS directive hazardous substances (phthalate esters and bromodiphenyl ethers) by novel mass spectrometry using soft ionization

Masataka Ohgaki, Yuko Takeguchi, Shin Okawa and Kenji Namiki

Hitachi High-Tech Science Corporation, Shintomi 2-15-5, Chuo-ku, Tokyo 104-0041, Japan

MO, 0000-0003-1930-9000

In 2013, the European Union expanded the list of hazardous substances contained in the RoHS Directive. In addition, certain phthalate esters, such as dibutyl phthalate (DBP), butyl benzyl phthalate (BBP), di-2-ethylhexyl phthalate (DEHP) and diisobutyl phthalate (DIBP), will be categorized as RoHS-prohibited substances starting from July 2019. Although pyrolysis–gas chromatography–mass spectrometry (Py-GC-MS) is a promising analytical method for the screening of phthalate esters, we have developed a novel soft-ionization MS method that is quantitative as well as faster and more convenient for this purpose. The sample was measured three times, after providing the calibration curve using a powdery standard material of SPEX. The data collection time is 5 min, and continuous measurements are completed within 8 min per sample. The mass spectrum was corrected by dividing the intensity by the sample weight. For the coefficient of variation, DBP was 2.9%, BBP was 3.4%, DEHP was 3.6%, and good reproducibility was observed. Precise analyses of phthalate esters using traditional methods can require solvent extraction times of up to 24 h as well as special techniques. Therefore, a screening method that can be easily carried out by anyone within ten minutes is very attractive.

## 1. Introduction

In Europe, RoHS Directive 2002/95/EC was issued in February 2003, [1] and the REACH regulation, which restricts the use of

six hazardous substances (Cd, Pb, Hg, Cr, polybrominated biphenyls and polybrominated diphenyl ethers), was implemented in July 2006 [2]. Furthermore, the RoHS Recast Directive (RoHS2, Directive 2011/65/EC) was issued in the *Official Journal of the European Union* in July 2011 [3]. According to this directive, the screening measurement of hazardous substances is mainly performed using X-ray fluorescence spectrometer, with which the measurement can be carried out non-destructively in about 10 min. If the results of screening measurement indicate a content close to regulated values, detailed measurement of the products is carried out to examine the content of specific hazardous substances. For the detailed measurement of polybrominated biphenyls and polybrominated diphenyl ethers, solvent extraction is carried out for more than 8 h as a pretreatment, and then the measurement is performed with a gas chromatograph–mass spectrometer (GC-MS).

In June 2015, four kinds of phthalate esters (PhEs), di-2-ethylhexyl phthalate (DEHP), butyl benzyl phthalate (BBP), dibutyl phthalate (DBP) and diisobutyl phthalate (DIBP), in addition to the regulated substances of the RoHS2 Directive, were added as target substances; the implementation will be from July 2019. As a result of the revision of the RoHS Directive, electrical and electronic equipment manufacturers are required to control, in a simple fast method, the content of PhEs in the products to be exported to Europe.

PhEs are used as a plasticizer for many resin products including polyvinyl chloride (PVC). The first successful use of plasticizer was in the preparation of xylonite resin, which was prepared in 1856 by Perkes by using camphor ($C_{10}H_{16}O$) as a plasticizer for nitrocellulose [4]. Hyatt also prepared the resin in 1868 by using camphor as a plasticizer for nitrocellulose and named it celluloid [5]. In 1933, Semon used high-boiling esters such as tricresyl phosphate as a plasticizer for PVC for the first time [6]. Some PhEs act on reproductive function, and their carcinogenicity is also suspected. The embryonic/reproductive toxicity was independently investigated by Poon *et al.* [7], Arcadi *et al.* [8] and Lamb *et al.* [9] and they showed that they are toxic. As for the hazards to environmental organisms, Rhodes *et al.* [10] and Call *et al.* [11] evaluated the toxicity to aquatic invertebrates, and Solyom *et al.* [12] evaluated the toxicity to amphibians.

For the measurement of PhEs, GC-MS is mainly used [13]. However, it is difficult to measure plastic products and food packaging materials as they are. Therefore, for high-precision measurement, Soxhlet extraction, which takes more than 6 h, or solvent extraction, which takes about 3 h, is carried out as a pretreatment. Then, GC-MS measurement (30–40 min) is carried out for the PhEs extracted from the products [14]. There is also a simple method in which a thermal desorption GC-MS apparatus is used. In the apparatus, a thermal extraction device is connected to the preceding GC-MS. Thus, part of the product (about 0.5 mg) is cut out and placed in a sample cup, and thermal extraction is carried out before measurement [15].

Plastic products contain various additives and volatile organic compounds; therefore, a high-resolution capillary column is used for GC to achieve a good separation. Because the amount of sample tends to become an overload for the capillary column, contaminants easily remain; thus a high frequency of column maintenance is necessary. To eliminate complicated operations and high running cost, a simple speedy GC-MS method has been sought [16].

As other measurement methods of PhEs, liquid chromatography with a UV detector (HPLC-UV) and Fourier-transform infrared spectrometry (FT-IR) may be considered; however, their application is difficult because the separation from contaminant compounds is not possible and the detection sensitivity is not satisfactory.

We developed a new apparatus, in which atmospheric pressure chemical ionization (APCI) excellent in selectivity, was used, and we are applying the apparatus to the present field [17]. Regarding the method using the direct mass spectrometry analysis method, there have been several reports so far, but none has been established as a simple and extremely rapid PhE screening analysis method [18–26]. Thus, a new measurement method of PhEs, without the use of a capillary column, was investigated. In the method, thermally vaporized PhEs are directly introduced, with nitrogen carrier gas, into the ionizing region of mass spectrometer and the ionization is achieved by APCI. We found that this method was very effective for a rapid screening analysis of PhEs [27].

As for the disputed brominated compounds, the determination of the species of organic bromine compounds cannot be achieved with a simple X-ray fluorescence spectrometer. However, it was found that the distinction of species of the regulated brominated flame retardants was possible with our newly developed apparatus, by which the distinction was achieved with the use of mass spectra, GC-MS measurement and the isotope ratio of brominated flame retardants.

Among the regulated brominated flame retardants, the widely used compounds are not highly toxic polybromobiphenyls. Less toxic polybromodiphenyl ethers (PBDEs) are most widely used as the flame

**Table 1.** Certified values for powder reference materials from SPEX CertiPrep.

| compound | CAS no. | purity (%) | certified (mg kg$^{-1}$) | uncertainty (mg kg$^{-1}$) |
|---|---|---|---|---|
| DBP | 84-74-2 | 99 | 998 | $\pm$ 120 |
| BBP | 85-68-7 | 98 | 994 | $\pm$ 119 |
| DEHP | 117-81-7 | 99 | 999 | $\pm$ 120 |

retardant for electrical products and fibres [28]. Among polybromodiphenyl ethers, the most used one is a decabrominated compound, namely decabromodiphenyl ether (decabromodiphenyl oxide, DBDE: CAS-No.1163-19-5). In the present study, the investigation was focused on this DBDE.

# 2. Material and methods

In this study, we conducted the following evaluations in order to apply our equipment: (i) investigation of the detectability of DBP, BBP, DEHP (detectable, how short you can measure, how to use the calibration curve method); (ii) DBP + DIBP, BBP, DEHP + DNOP were prepared in each PE matrix, and the effectiveness of the apparatus for analysis of these samples was evaluated; (iii) DBP, BBP, DEHP: we made samples containing 500, 1000, 1500 and 2000 $\mu$g g$^{-1}$ each in PVC and investigated the correlation between analysis results and concentration values by our equipment; (iv) investigation of the extensibility of our analysis subjects. (Here, because bromodiphenyl ether is also subject to regulation in RoHS2, we investigated the detection of these substances.)

## 2.1. Measurement of PhEs

To confirm that the detection of PhEs is sufficiently possible by our method, a mixed sample of four kinds of PhEs, DIBP, DBP, BBP and DEHP, was analysed. The mixed sample was diluted and an appropriate amount was weighed; in addition, an appropriate amount of sample containing PhEs was also weighed, and they were placed in the autosampler. A high-precision electronic balance (MSE2.7S, Sartorius) was used for weighing samples.

When the samples placed in the autosampler are introduced into the furnace, the contained PhEs vaporize, and the vaporized material is introduced into the mass analysis section with nitrogen carrier gas. On this occasion, protonated ions are formed by APCI with corona discharge, and the quantification is achieved by mass separation. We named this system 'a thermal desorption mass spectrometer' (Thermal Desorption MS). (The present product name is HM1000A, Hitachi High-Tech Science Corporation.) Thermal extraction conditions and mass measurement conditions are shown below.

Thermal extraction conditions: thermal extraction temperature (specimen stage temperature): 80–230°C, atmospheric gas: nitrogen 1 l min$^{-1}$, furnace temperature: 300°C, tube temperature: 330°C, measurement cycle time: about 10 min.

MS measurement conditions: ionization method: APCI, AP1 temperature: 170°C, mode: SIM.

## 2.2. Calibration curves for PhEs

To confirm the reliability of this apparatus for the concentration measurement of PhEs, the verification experiment was carried out with the use of powder reference materials from SPEX CertiPrep. The certified values for the used reference materials are shown in table 1. The respective reference materials (0.2 mg) were weighed, and the concentration measurement of PhEs was carried out with the thermal desorption MS apparatus. For the measurement, similar conditions to those of §2.1 were used. Based on the certified values for the respective samples, calibration curves to determine the concentration were prepared, and the effectiveness of this measurement method was examined figure 1.

## 2.3. Measurement of bromodiphenyl ethers

As the standard sample for the brominated compound DBDE, which is most widely used as a flame retardant for electrical products and fibres, CRM8110a by the National Metrology Institute of Japan of the National Institute of Advanced Industrial Science and Technology was used.

R. Soc. open sci. **6**: 181469

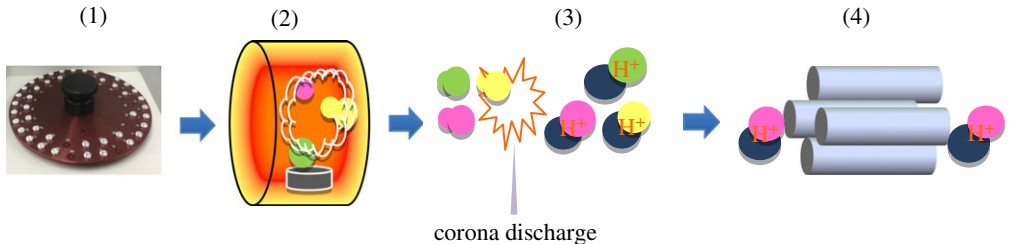

**Figure 1.** Flowchart of analytical method of thermal desorption mass spectra using soft ionization method.

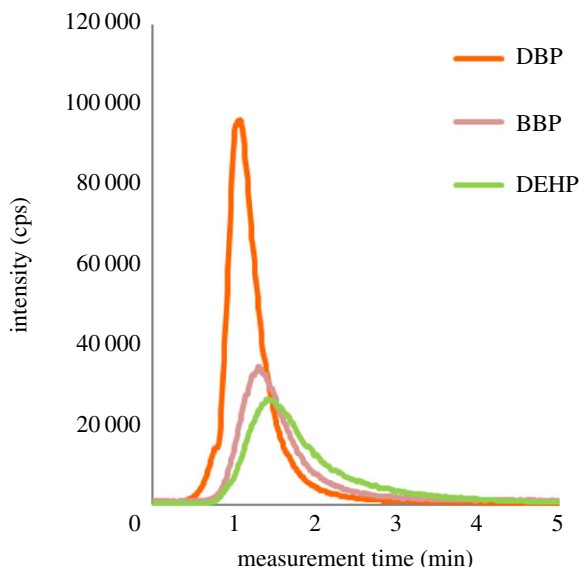

**Figure 2.** Thermal desorption peak profile of SIM mode measurement.

For the measurement of bromodiphenyl ethers, as is the case with the PhEs, 0.2 mg of the sample was weighed and placed in the autosampler. The sample was heated in the furnace and the bromodiphenyl ethers were vaporized. The bromodiphenyl ethers introduced into the mass analysis section were ionized by APCI. Scan mode was used for measurement. Thermal extraction conditions and mass analysis conditions are shown below.

Thermal extraction conditions: thermal extraction temperature (sample stage temperature): $80-230°C$ (90 s)$-310°C$ (360 s), atmospheric gas: nitrogen $1\,l\,min^{-1}$, furnace temperature: $300°C$, tube temperature: $330°C$, measurement cycle time: about 15 min.

MS measurement conditions: ionization method: APCI, AP1 temperature: $170°C$, mode: scan.

# 3. Results and discussion

## 3.1. Calibration curve for PhEs

Mass spectra of PhEs measured in SIM mode are shown in figure 2. As seen in this figure, the data collection time was about 7 min. In the continuous measurement, the analysis can easily be completed in 10 min per sample including the time for sample change; the measurement speed is clearly fast. The isomers DBP and DIBP have identical mass numbers. As is the case with the ion attachment mass spectrometry (IA-MS), [29–31] the mass separation cannot directly be achieved, with the present system, for the isomers with identical mass numbers such as DBP and DIBP (figure 3). Accordingly, DBP and DIBP were evaluated as one mass spectrum of the same mass number (table 2).

## 3.2. Mixed samples of PhEs and polyethylene (PE) resin

According to the prescribed method of screening analysis of PhEs by Py/TD-GC-MS described in the International Standard of IEC 62321-8, a blank sample is used for verification of 'Check contamination

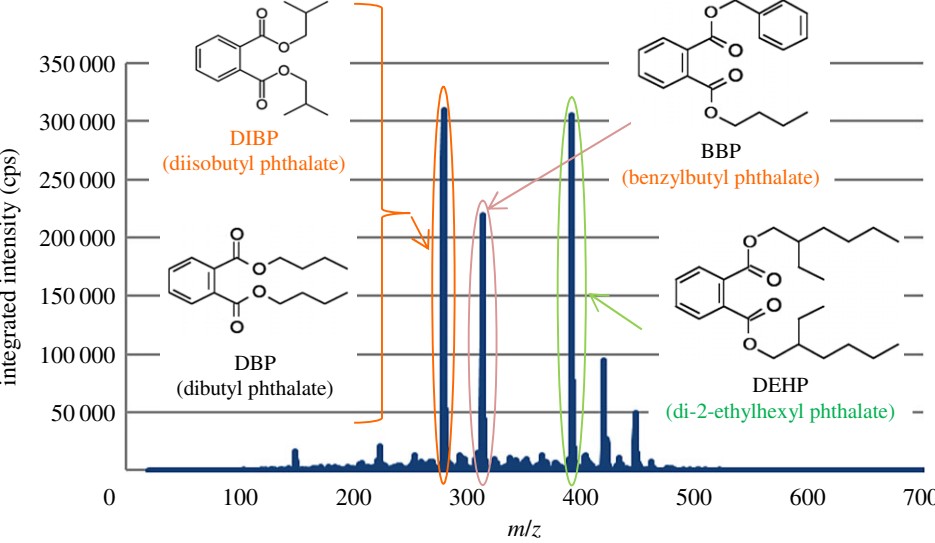

**Figure 3.** APCI mass spectra of PHE standard.

**Table 2.** Certified value for CRM8110a reference material (by the National Metrology Institute of Japan of the National Institute of Advanced Industrial Science and Technology).

| compound | CAS no. | certified (mg kg$^{-1}$) | uncertainty (mg kg$^{-1}$) |
|---|---|---|---|
| DBDE | 1163-19-5 | 886 | $\pm$ 28 |

and carry over', 100 mg kg$^{-1}$ standard substance is used for 'Check the sensitivity' verification, and 1000 mg kg$^{-1}$ standard substance is used for 'Calibration' [1]. Therefore, it is standardized to screen and analyse 500–2000 mg kg$^{-1}$ PhEs using this 1000 mg kg$^{-1}$ standard substance. According to this method, in this study 1000 mg kg$^{-1}$ standard substance was incorporated into the process as a calibration curve.

Certified values for concentrations of the respective PhEs in the powder reference materials from SPEX CertiPrep are shown in table 3. Calibration curves obtained by using these reference materials are shown in figure 4.

We prepared our own samples by mixing seven kinds of PhEs, namely DBP, BBP, DEHP, DIBP, diisodecyl phthalate (DIDP), diisononyl phthalate (DINP) or di-n-octyl phthalate (DNOP) with PE so that the respective concentrations were 1000 mg kg$^{-1}$. In addition, we prepared PE resin evaluation samples containing three kinds of PhEs, DBP + DIBP, BBP or DEHP + DNOP, by using our own samples described above. The concentrations and mass numbers ($m/z$) of the respective PhEs, for the prepared three samples, are shown in table 4.

The above three samples were analysed by our new screening analysis method. Because DEHP and DNOP have identical mass numbers as stated above, the combined amount was spectrally evaluated. The calibration curves shown in figure 4 were used for evaluation, and the measurement was repeated six times for the respective samples. The obtained results of quantitative analysis as well as the repeatability (CV value) are shown in table 5.

The recovery rates of the respective PhEs, which were obtained by dividing the mean analysis values by the loaded concentrations of PhEs, are also shown in table 5. The obtained repeatability for the respective PhEs (DBP/DIBP: 6.9%, BBP: 6.3%, and DEHP/DNOP: 8.7%) was less than 10%, and remarkably good results were obtained as a screening analysis.

The deviation of the quantitative values from the concentrations of the loaded PhEs was 6.9% for DBP/DIBP, 6.3% for BBP and 8.7% for DEHP/DNOP. Thus, the results sufficiently satisfied the criterion 'within 30%' described in the detailed analysis method of PhEs (IEC-62321-8). Furthermore, the mean recovery rates (DBP/DIBP: 91.5%, BBP: 96.7% and DEHP/DNOP: 83.8%) were good, and the satisfactory accuracy was obtained.

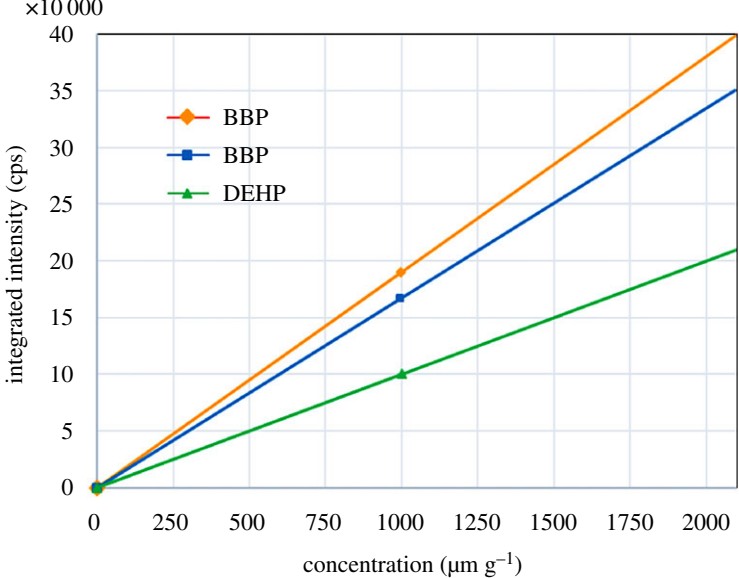

**Figure 4.** Calibration curves prepared for DBP, BBP and DEHP by using powder reference materials from SPEX CertiPrep.

**Table 3.** Certified values for powder reference materials from SPEX CertiPrep.

| reference material | CAS no. | purity | certified | uncertainty |
|---|---|---|---|---|
| DBP | 84-74-2 | 99% | 998 $\mu g\ g^{-1}$ | $\pm$ 120 $\mu g\ g^{-1}$ |
| BBP | 85-68-7 | 98% | 994 $\mu g\ g^{-1}$ | $\pm$ 119 $\mu g\ g^{-1}$ |
| DEHP | 117-81-7 | 99% | 999 $\mu g\ g^{-1}$ | $\pm$ 120 $\mu g\ g^{-1}$ |

**Table 4.** Amounts of loaded PhEs in polyethylene (PE) resin for the mixed samples.

| compound | concentration | weight | m/z (protonated) |
|---|---|---|---|
| DBP + DIBP | 2101 $\mu g\ g^{-1}$ | DBP: 1.598 g | 279.3 |
| | | DIBP: 1.552 g | |
| BBP | 1030 $\mu g\ g^{-1}$ | 1.546 g | 313.4 |
| DEHP + DNOP | 2072 $\mu g\ g^{-1}$ | DEHP: 1.556 g | 391.6 |
| | | DNOP: 1.553 g | |

## 3.3. Range of quantification for PhEs

The regulatory concentration of PhEs is 1000 mg kg$^{-1}$ in RoHS2 [3]. The preparation method of phthalate standard samples described in IEC62321-8, which was issued in March 2017, was followed to prepare evaluation samples. PVC resin was diluted with tetrahydrofuran (THF), and then PhEs were added so that the concentration was 500, 1000, 1500 and 2000 mg kg$^{-1}$. The concentration of the PhEs was analysed for these samples, and the range of quantification was examined. The results of concentration measurement are shown in table 6. The obtained calibration curves for the respective PhEs are shown in figure 5. As shown in the figure, the linearity between the concentrations of the respective PhEs and the analysed concentrations was high, and the correlation coefficient was higher than 0.990; thus good calibration curves were obtained. Furthermore, the agreement of both concentrations near the regulatory concentration 1000 mg kg$^{-1}$ was high and accurate. Thus, the application of this screening analysis is highly possible at the regulatory concentration.

## 3.4. Examination of the measurement of bromodiphenyl ethers

Bromodiphenyl ethers regulated by RoHS2 contain bromine elements; therefore, if the natural isotope ratio of bromine element can be measured with our newly developed Thermal Desorption MS, the

**Table 5.** Results of quantitative analysis, repeatability and the recovery rates for the mixed samples of polyethylene (PE) resins containing PhEs.

| | DBP + DIBP ($\mu$g g$^{-1}$) | BBP ($\mu$g g$^{-1}$) | DEHP + DNOP ($\mu$g g$^{-1}$) |
|---|---|---|---|
| 1 | 1997 | 1034 | 1822 |
| 2 | 1785 | 959 | 1582 |
| 3 | 1967 | 982 | 1763 |
| 4 | 1749 | 914 | 1544 |
| 5 | 2101 | 1094 | 1949 |
| 6 | 1931 | 991 | 1756 |
| average | 1922 | 996 | 1736 |
| RD | 133 | 62 | 151 |
| CV | 6.9% | 6.3% | 8.7% |
| recovery | 91.5% | 96.7% | 83.8% |

**Table 6.** Measurement results for the standard solutions when the range of quantification of PhEs was examined.

| | analysed concentration | | |
|---|---|---|---|
| prep. conc. | DBP | BBP | DEHP |
| 0 | 59.0 | 25.0 | 37.8 |
| 0 | 28.2 | 37.5 | 50.4 |
| 500 | 469.3 | 471.2 | 509.9 |
| 500 | 481.4 | 474.0 | 490.0 |
| 1000 | 949.1 | 945.4 | 940.8 |
| 1000 | 951.8 | 944.1 | 951.1 |
| 1500 | 1392.8 | 1428.4 | 1478.7 |
| 1500 | 1370.7 | 1438.6 | 1537.2 |
| 2000 | 1845.8 | 1944.0 | 2136.3 |
| 2000 | 1767.7 | 1952.9 | 2090.3 |

speedy distinction of the compounds is possible. We carried out speedy screening measurement of brominated compounds with this apparatus, and we could identify bromodiphenyl ethers within only 15 min; we obtained a clear bromine spectrum pattern with high agreement (figure 6).

The distinction of bromodiphenyl ethers can be achieved in the following way. The regulated bromodiphenyl ethers contain bromine elements; therefore, the distinction of the compounds can be carried out fast by measuring the natural isotope ratio of bromine element with a mass spectrometer. For a molecule containing $n$ bromine elements, the calculation is performed with the binomial distribution formula in the following way:

$$P_{n,k} = \frac{1}{A} \binom{n}{k} p^k (1-p)^{n-k} = \frac{1}{A} \frac{n!}{k!(n-k)!} p^k (1-p)^{n-k}.$$

Here, $A$ is a normalization factor.

$$A = \sum_{k=1}^{n} P_{n,k} = \sum_{k=1}^{n} \frac{n!}{k!(n-k)!} p^k (1-p)^{n-k}.$$

The symbol $p$ is the relative abundance 0.4931 of bromine isotope $^{81}$Br. In the case of DBDE, $n = 10$, and the relative abundance of DBDE can be calculated from the number of the isotope and the precise mass number; they are shown in table 7.

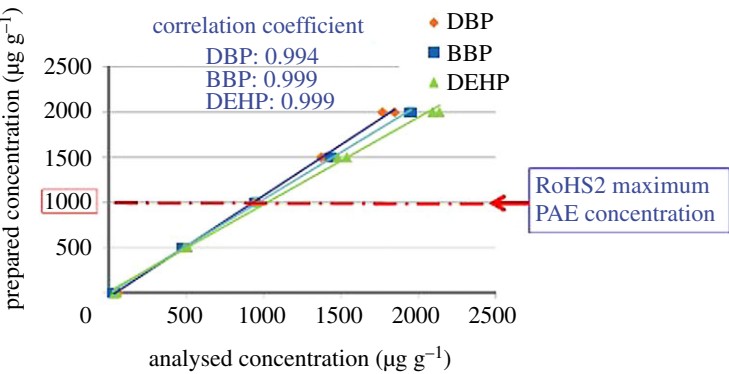

**Figure 5.** Correlation between the concentrations of DBP, BBP and DEHP in the prepared samples and the respective concentrations obtained by our new analysis method.

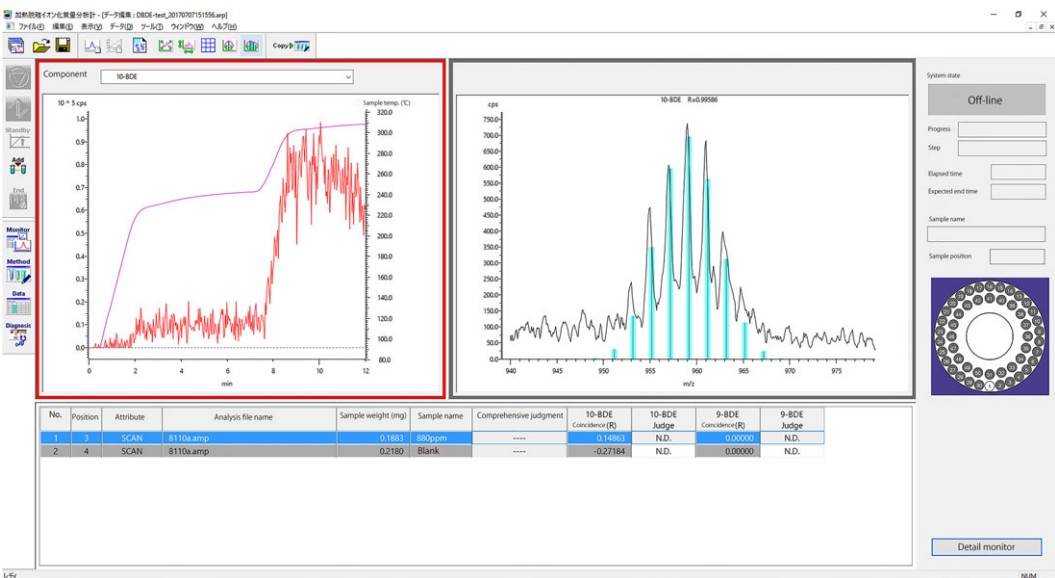

**Figure 6.** Measurement example of bromodiphenyl ethers.

The correlation coefficient (R), which is the indicator for the distinction of compounds, is calculated using the following equation:

$$R = \frac{\sum_{i=1}^{n}(x_i - \bar{x})(y_i - \bar{y})}{\sqrt{\sum_{i=1}^{n}(x_i - \bar{x})^2}\sqrt{\sum_{i=1}^{n}(y_i - \bar{y})^2}}.$$

Here, $x_i$, $y_i$, $\bar{x}$ and $\bar{y}$ represent the theoretical intensity, measured intensity, mean theoretical intensity and mean measured intensity, respectively.

In this study, a polystyrene standard sample containing 1000 mg kg$^{-1}$ of DBDE was actually measured, and the correlation coefficient (R) was found to be 0.9998. The spectrum of the polystyrene standard sample containing 1000 mg kg$^{-1}$ of DBDE is shown with the correlation coefficient in figure 7.

Because compounds with high boiling points are contained in the sample of DBDE etc. the necessary thermal desorption time is about 5 min longer than that for PhEs; as a result, the measurement time becomes that much longer. However, the quantitative analysis of this sample took more than 12 h with the past high-precision analysis method. Therefore, the present method, in which the analysis of one sample can be completed in about 15 min, is a very rapid and remarkably effective screening analysis.

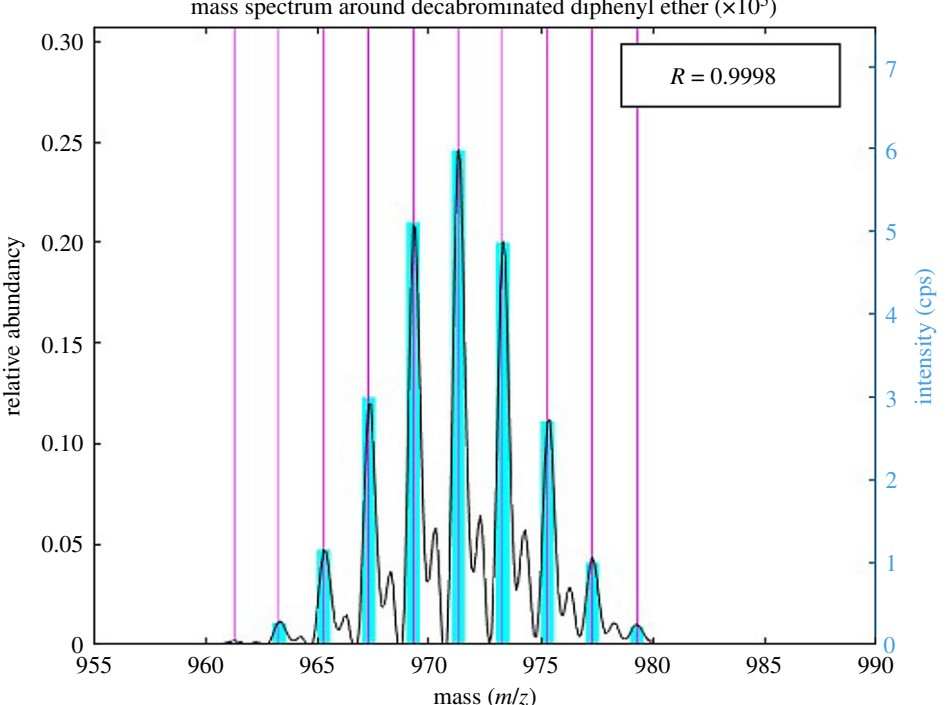

**Figure 7.** Spectrum of polystyrene standard sample containing 1000 mg kg$^{-1}$ of DBDE and the correlation coefficient ($R$).

**Table 7.** Number of $^{81}$Br, precise mass number and relative abundance of DBDE.

| number of $^{81}$Br | precise mass number | relative abundance of DBDE |
|---|---|---|
| 0 | 949.2 | 0.0046 |
| 1 | 951.2 | 0.0443 |
| 2 | 953.2 | 0.1940 |
| 3 | 955.2 | 0.5032 |
| 4 | 957.2 | 0.8567 |
| 5 | 959.2 | 1.0000 |
| 6 | 961.2 | 0.8106 |
| 7 | 963.2 | 0.4506 |
| 8 | 965.2 | 0.1644 |
| 9 | 967.2 | 0.0355 |
| 10 | 969.2 | 0.0035 |

## 3.5. Verification of effectiveness as an apparatus—inspection of the performance-based measurement system (PBMS, IEC 62321-1)

The verification that our newly developed thermal desorption MS system with soft ionization is a reliable apparatus was carried out as follows. Performance-based Analytical Test Method Approach means a measurement system based upon established performance criteria for accuracy and precision with the use of analytical test methods. As used in this subsection, this is a measurement system used by laboratories to demonstrate that a particular analytical test method is acceptable for demonstrating compliance. We investigated the performance of our newly developed analysis device.

The accuracy and precision of the measured values by the present apparatus were verified, and the validation was carried out. The accuracy was studied by using the standard sample NMIJ CRM8152-a. Samples of DBP/DIBP: 1835 mg kg$^{-1}$, BBP: 897 mg kg$^{-1}$ and DEHP/DNOP: 1836 mg kg$^{-1}$ were repeatedly measured 10 times, and the certificate of accuracy (COA) was evaluated. As for method

**Table 8.** Measurement of NMIJ CRM8152-a standard sample.

|  | DBP 1835 mg kg$^{-1}$ | BBP 897 mg kg$^{-1}$ | DEHP 1836 mg kg$^{-1}$ |
|---|---|---|---|
| 1 | 1804 | 881 | 1847 |
| 2 | 2052 | 975 | 1786 |
| 3 | 1997 | 954 | 1846 |
| 4 | 2090 | 1002 | 1925 |
| 5 | 1699 | 845 | 1866 |
| 6 | 1722 | 859 | 1839 |
| 7 | 1703 | 831 | 1888 |
| 8 | 1791 | 877 | 1928 |
| 9 | 1580 | 827 | 1834 |
| 10 | 1930 | 932 | 1882 |
| average | 1837 | 898 | 1864 |
| COA | 2 | 1 | 28 |
| s.d. | 171 | 63 | 43 |
| CV% | 9.3% | 7.0% | 2.3% |

**Table 9.** Measurement of dried 500 mg kg$^{-1}$ solution.

| no. | DBP mg kg$^{-1}$ | BBP mg kg$^{-1}$ | DEHP mg kg$^{-1}$ | no. | DBP mg kg$^{-1}$ | BBP mg kg$^{-1}$ | DEHP mg kg$^{-1}$ |
|---|---|---|---|---|---|---|---|
| 1 | 497 | 550 | 552 | 13 | 524 | 546 | 543 |
| 2 | 492 | 531 | 529 | 14 | 465 | 478 | 475 |
| 3 | 475 | 504 | 504 | 15 | 445 | 451 | 455 |
| 4 | 498 | 523 | 527 | 16 | 431 | 438 | 435 |
| 5 | 530 | 562 | 563 | 17 | 446 | 448 | 453 |
| 6 | 435 | 462 | 491 | 18 | 474 | 481 | 483 |
| 7 | 489 | 507 | 506 | 19 | 432 | 439 | 438 |
| 8 | 488 | 504 | 508 | 20 | 518 | 530 | 524 |
| 9 | 485 | 503 | 506 | average | 479 | 498 | 500 |
| 10 | 484 | 502 | 497 | COA | 21 | 2 | 0 |
| 11 | 481 | 500 | 496 | s.d. | 29 | 37 | 36 |
| 12 | 482 | 508 | 510 | CV% | 6.1 | 7.4 | 7.1 |

detection limit (MDL), samples of dried 500 mg kg$^{-1}$ solutions were repeatedly measured 20 times, and the detection limit was evaluated from their standard deviations (s.d.s). As for the validation, if the accuracy test results meet the criteria COA $\leq$ 15% and CV $\leq$ 15%, they were marked 'Pass', and if MDL meets the criteria $\leq$ 100 mg kg$^{-1}$, they were marked 'Pass', and the obtained respective parameters were examined as the comprehensive evaluation (tables 8–10).

As seen in the tables, the accuracies obtained in the measurement of DBP/DIBP, BBP and DEHP/DNOP were as follows: DBP/DIBP: 2 mg kg$^{-1}$ for 1835 mg kg$^{-1}$, BBP: 1 mg kg$^{-1}$ for 897 mg kg$^{-1}$ and DEHP/DNOP: 45 mg kg$^{-1}$ for 1836 mg kg$^{-1}$, and their accuracies were very high. In addition, the CV values were 9.3%, 7.0%, and 2.3%, respectively, and they sufficiently satisfied the criterion '15% or less'. As seen in the total evaluation of validation parameters, 'Pass' mark was obtained in all the categories, and this newly developed system was demonstrated to be a satisfactory analysis apparatus.

**Table 10.** Example of validation parameter.

| accuracy/repeatability (criteria: COA ≦ 15%, CV ≦ 15%) | | | | | | | | | MDL (PhEs) (criteria: ≦ 100 mg kg⁻¹) | | | MDL (DBDE) |
|---|---|---|---|---|---|---|---|---|---|---|---|---|
| DBP/DIBP: 1835 mg kg⁻¹ | | | BBP: 897 mg kg⁻¹ | | | DEHP/DNOP: 1836 mg kg⁻¹ | | | DBP/DIBP: 1835 mg kg⁻¹ | BBP: 897 mg kg⁻¹ | DEHP/DNOP: 1836 mg kg⁻¹ | DBDE: 836 mg kg⁻¹ |
| average (mg kg⁻¹) | CV% | COA (mg kg⁻¹) | average (mg kg⁻¹) | CV% | COA (mg kg⁻¹) | average (mg kg⁻¹) | CV% | COA (mg kg⁻¹) | MDL (mg kg⁻¹) | MDL (mg kg⁻¹) | MDL (mg kg⁻¹) | MDL (mg kg⁻¹) |
| 1817 | 6.16 | 56 | 962 | 4.12 | 58 | 1805 | 7.07 | 62 | 8.5 | 12.6 | 25.9 | 63.9 |
| | pass | pass | | pass | pass | | pass | pass | pass | pass | pass | pass |

# 4. Conclusion

Our newly developed thermal desorption MS system with soft ionization has the following characteristics as a screening apparatus.

— The system was developed based on the novel idea that protonated ions are formed by APCI and the quantification is achieved by mass separation.
— High-precision quantitative analysis can be achieved though it is a screening analysis system.
— The analysis is suitable for PhEs.
— The system is effective for the quantitative analysis of brominated compounds.
— Extremely speedy measurement is possible.

Thus, the method devised in this study is expected to be very useful, for an urgently needed massive analysis of PhEs, to respond to the requirement of RoHS2.

With the implementation of the RoHS2 from July 2019, testing needs for PhEs are expected to increase; however, it is necessary to initially determine whether PhEs are contained in the products by simple fast measurement with a screening apparatus. In the system created in this study, the analysis of PhEs can be achieved in about 10 min by only placing about 0.2 mg of a sample on the sample pan. We also studied the automation and developed a turntable on which 50 samples can be continuously measured. Thus, the continuous measurement for about 8 h and automatic screening measurement are possible; therefore, it is a highly effective system for screening analysis.

In the traditional analysis method for phthalates, GC-MS is used and helium gas is used as the carrier gas. Therefore, the troublesome supply and management of helium cylinders are necessary. In our present system, helium gas is not used and the measurement is rapid; thus the running cost can be lowered. Thus, the system can be effectively applied to the analysis of PhEs to respond to the requirement of RoHS2, and high productivity in analysis can be expected.

Data accessibility. This paper is completed within the paper including various data.
Authors' contributions. M.O. contributed to the analysis and interpretation of data and final approval of the version to be published. Y.T. contributed to the acquisition of data. S.O. contributed to the acquisition of data and drafting the article or revising critically. K.N. contributed to the analysis of data.
Competing interests. We have no competing interests.
Funding. The research activity fund of all researchers is made by our company Hitachi High-Tech Science Corporation.
Acknowledgements. We gratefully acknowledge the work of pilot model equipment production members Mr Watanabe, Mr Akiyama and Mr Sakai of engineering team in our corporation. We also thank Mr Matoba of market division member, for drafting the plan of the equipment.

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
