## [Reviewer comments · Royal Society Open Science]

Review History

RSOS-181469.R0 (Original submission)

Review form: Reviewer 1 (Haiyang Li)

Is the manuscript scientifically sound in its present form?

Yes

Are the interpretations and conclusions justified by the results?

No

Is the language acceptable?

Yes

Is it clear how to access all supporting data?

Not Applicable

Do you have any ethical concerns with this paper?

No

Have you any concerns about statistical analyses in this paper?

I do not feel qualified to assess the statistics

Recommendation?

Major revision is needed (please make suggestions in comments)

Comments to the Author(s)

This paper introduce a soft ionization mass spectrometry method for rapid measurement of phthalate ester and bromodiphenyl ether, which is possible be used for screening analysis of RoHS. This paper could be accepted for publication after the following revision. Major concerns: (1) the method should be verified by real samples, how about the matrix effect? whether other hazardous substances, such as VOCs and SVOCs affect the quantitation of PAEs? (2) The data points for calibration curves shown in figure 4 is too small to get a reliable fitting; (3) what is relationship between the calibration curves shown in figure 4 and figure 5.

Minor concerns: (1) the author should cite more literature papers on direct mass spectrometry analysis method using ambient ionization method; (2) Figure 7 and Figure 8 are not necessary, as the isotope pattern agreements tell nothing but the molecular assignments

Review form: Reviewer 2 (Nagegownivari Ramachandra Munirathnam)

Is the manuscript scientifically sound in its present form?

Yes

Are the interpretations and conclusions justified by the results?

Yes

Is the language acceptable?

Yes

Is it clear how to access all supporting data?

Yes

Do you have any ethical concerns with this paper?

No

Have you any concerns about statistical analyses in this paper?

No

Recommendation?

Accept with minor revision (please list in comments)

Comments to the Author(s)

Given separately in a file (Appendix A).

Decision letter (RSOS-181469.R0)

06-Nov-2018

Dear Dr Ohgaki:

Title: Screening analysis of RoHS Directive hazardous substances (Phthalate esters and Bromodiphenyl ether) by novel mass spectrometry using soft ionization
Manuscript ID: RSOS-181469

The editor assigned to your manuscript has now received comments from reviewers. We would like you to revise your paper in accordance with the referee and Subject Editor suggestions which can be found below (not including confidential reports to the Editor). Please note this decision does not guarantee eventual acceptance.

Please submit your revised paper before 29-Nov-2018. Please note that the revision deadline will expire at 00.00am on this date. If we do not hear from you within this time then it will be assumed that the paper has been withdrawn. In exceptional circumstances, extensions may be possible if agreed with the Editorial Office in advance. We do not allow multiple rounds of revision so we urge you to make every effort to fully address all of the comments at this stage. If deemed necessary by the Editors, your manuscript will be sent back to one or more of the original reviewers for assessment. If the original reviewers are not available we may invite new reviewers.

RSC Associate Editor:
Comments to the Author:
(There are no comments.)

RSC Subject Editor:
Comments to the Author:
(There are no comments.)

Reviewers' Comments to Author:
Reviewer: 1

Comments to the Author(s)

This paper introduce a soft ionization mass spectrometry method for rapid measurement of phthalate ester and bromodiphenyl ether, which is possible be used for screening analysis of RoHS. This paper could be accepted for publication after the following revision. Major concerns: (1) the method should be verified by real samples, how about the matrix effect? whether other hazardous substances, such as VOCs and SVOCs affect the quantitation of PAEs? (2) The data points for calibration curves shown in figure 4 is too small to get a reliable fitting; (3) what is relationship between the calibration curves shown in figure 4 and figure 5.

Minor concerns: (1) the author should cite more literature papers on direct mass spectrometry analysis method using ambient ionization method; (2) Figure 7 and Figure 8 are not necessary, as the isotope pattern agreements tell nothing but the molecular assignments

Reviewer: 2

Comments to the Author(s)
Given separately in a file

Author's Response to Decision Letter for (RSOS-181469.R0)

See Appendices B & C.

Decision letter (RSOS-181469.R1)

18-Dec-2018

Dear Dr Ohgaki:

Title: Screening analysis of RoHS Directive hazardous substances (Phthalate esters and Bromodiphenyl ether) by novel mass spectrometry using soft ionization
Manuscript ID: RSOS-181469.R1

It is a pleasure to accept your manuscript in its current form for publication in Royal Society Open Science. The chemistry content of Royal Society Open Science is published in collaboration with the Royal Society of Chemistry.

RSC Associate Editor
Comments to the Author:
(There are no comments.)

Reviewer(s)' Comments to Author:

Appendix A

Point1: Certified values for concentrations of the respective PhEs in the powder reference materials from SPEX CertiPrep are shown in Table 3. Calibration curves obtained by using these reference materials are shown in Fig. 4.

Query 1: As per Table -3 the certified reference material values are around 1000 mg/kg (ppm) for DIBP, BBP and DEHP but in Fig.4 the calibration range of above standards are 250 to 1500 $\mu\text{g/g}$ (ppm). Reasoning is necessary.

Point 2. The above three samples were analyzed by our new screening analysis method. Because DEHP and DNOP have identical mass numbers as stated above, the combined amount was spectrally evaluated. The calibration curves shown in Fig. 4 were used for evaluation, and the measurement was repeated 6 times for the respective samples. The obtained results of quantitative analysis as well as the repeatability (CV value) are shown in Table 5.

Query 2: In Fig.4 the calibration curve range 250 to 1500 $\mu\text{g/g}$ (ppm). But in the Table.5 the repeatability results are more than calibration range. As per quantification rule the results are within the calibration range.

Query 3: In Fig.2 shows the Thermal desorption peak profile of SIM mode measurement of DBP, BBP & DEHP. But there is no Thermal desorption peak of DBDE observed.

Query 3: In Fig.4 the calibration curves of DBP, BBP & DEHP prepared using powder reference material. But there is no calibration a curve of DBDE was observed even though the reference material was available. (As you mentioned in manuscript).

Needs reasoning and explanation of the above points.

Appendix B

Manuscript ID: RSOS-181469

Thank you very much for helping us review the paper entitled “Screening analysis of RoHS Directive hazardous substances (Phthalate esters and Bromodiphenyl ether) by novel mass spectrometry using soft ionization.”

We are submitting this revised manuscript after considering the suggestions made by two reviewers. Answer to the respective reviewers and major changes we have made are follows.

Response to Reviewer: 1

Answer for major concerns:

(1) the method should be verified by real samples, how about the matrix effect? whether other hazardous substances, such as VOCs and SVOCs affect the quantitation of PAEs?

→ As you say, this method requires verification on actual samples, and we are currently conducting measurements on some samples. However, since the purpose of this paper is to evaluate whether or not the developed apparatus can detect phthalate esters to a minimum, we verified by using two popular samples with known concentrations.

We have to consider the effect of the matrix as you say (that is, the difference in accuracy of measurement data due to matrix material). Also, VOCs and SVOCs are also a big issue, and some SVOCs contain PhEs in particular, and they are considered as the target application of this system. In this paper, we kept in mind that we urgently announce the equipment that can apply to the new RoHS2. In the paper, as a representative sample, we made a sample containing PhEs in polyethylene matrix. We used this sample to evaluate the analytical performance of the apparatus (the second paragraph of section 4-2). In addition, we prepared samples containing several concentrations of PhEs in PVC and verified the quantitative range for screening analysis (section 4-3).

We will also verify the versatility by acquiring the data on the effects of various other matrix substances, so please wait for the paper that we will write in the near future.

(2) The data points for calibration curves shown in figure 4 is too small to get a reliable fitting

→ As you know, fitting with several specimens with different concentrations makes it possible to perform highly reliable quantitative analysis. However, the aim of this paper is to verify the equipment related to screening analysis aimed at rapid analysis.

Since the probability in the vicinity of the PhE concentration of 1,000 µg/g, which is the target of the threshold in RoHS2 screening analysis, is the most important, Fig.4 simply shows the calibration curve obtained with 1000 µg/g PhEs concentration of SPEX standard substance. Of course, in case of detailed quantitative analysis a calibration curve consisting of two points is insufficient, but from the viewpoint of rapid screening analysis it was used.

The most important point that you say is that the axis range is different from the actual calibration range, and I agree with you that it is difficult to recognize using the standard substance in Table 3. In order to clarify the standard substance concentration and calibration range used in this research, we have to replace Fig.4.

Also, the rest are supplementary. Regarding the method of screening analysis of RoHS2 using standard

substance of only 1000 µg/g concentration, it is described as the method prescribed in "Screening analysis of PhEs by Py/DT-GC-MS" in the International standard document of IEC 62321 – 8. And I also experimented with those method in this paper. Then, I also added to the text of this paper about what I mentioned here.

(3) what is relationship between the calibration curves shown in figure 4 and figure 5.

→ Although we mentioned a little more in the above, Fig.4 is a calibration curve prepared using SPEX's powder standard substance. Simply verified whether DBP, BBP, DEHP containing substances in the vicinity of the target concentration can be detected (described at first line at 4-2). Furthermore, we prepared a sample for evaluation in which PhEs with concentrations of 500, 1000, 1500 and 2000 µg/g were contained in the PVC resin matrix. Fig.5 shows the correlation between each analytical value obtained by our apparatus and the actual concentration. (Described in 4-3). The notation of "Calibration curve" is not suitable for this content, so I changed the caption of the figure.

Answer for minor concerns:

(1) the author should cite more literature papers on direct mass spectrometry analysis method using ambient ionization method;

→ Surely it is. We have increased cited references of direct mass spectroscopy analysis, a technique relatively close to the method of this paper, which was used for reference when developing the paper content.

(2) Figure 7 and Figure 8 are not necessary, as the isotope pattern agreements tell nothing but the molecular assignments

→ Thank you very much for your suggestion. As you say, Fig.7 concludes that it is unnecessary because it only spectrally patterns the number of ⁸¹Br, precise mass number, and relative abundance of DBDE. On the other hand, Fig.8 shows the meaningful correlation (coincidence) between the spectral pattern obtained by measuring the actual DBDE-containing polystyrene standard substance and the theoretical spectrum pattern for discriminating the abundance ratio of DBDE, and Fig.8 proves that the spectrum of the actual DBDE standard is highly correlated with the spectra obtained by discriminating compounds by measuring isotopic ratios of bromine.

As a result of considering the above, Fig. 7 was deleted according to your advice.

Appendix C

Manuscript ID: RSOS-181469

Thank you very much for helping us review the paper entitled "Screening analysis of RoHS Directive hazardous substances (Phthalate esters and Bromodiphenyl ether) by novel mass spectrometry using soft ionization."

We are submitting this revised manuscript after considering the suggestions made by two reviewers. Answer to the respective reviewers and major changes we have made are follows.

Response to Reviewer: 2

Point1: Certified values for concentrations of the respective PhEs in the powder reference materials from SPEX CertiPrep are shown in Table 3. Calibration curves obtained by using these reference materials are shown in Fig. 4.

Query 1: As per Table -3 the certified reference material values are around 1000 mg/kg (ppm) for DIBP, BBP and DEHP but in Fig.4 the calibration range of above standards are 250 to 1500 $\mu\text{g/g}$ (ppm).

Reasoning is necessary.

→ Fig.4 is a calibration curve prepared using SPEX's powder standard substance (Table 3). Simply verified whether DBP, BBP, DEHP containing substances in the vicinity of the target concentration can be detected (described in the first line of 4-2). However, as you recognized, it is difficult to recognize using the standard substance in Table 3. It does not reflect the calibration range actually used and it might be difficult to understand, therefore, Fig.4 was replaced in order to clarify that we used the concentration using the standard substance listed in Table 3 and calibration range.

On the other hand, we also prepared samples for evaluation in which PhEs with concentrations of 500, 1000, 1500 and 2000 $\mu\text{g/g}$ were obtained in PVC resin matrix. Fig.5 shows the correlation between each analytical value of these samples and the actual concentration (as described in 4-3). Also, because the notation of "Calibration curve" is not suitable for this content, we changed the figure caption to "Correlation between ...".

Point 2. The above three samples were analyzed by our new screening analysis method. Because DEHP and DNOP have identical mass numbers as stated above, the combined amount was spectrally evaluated. The calibration curves shown in Fig. 4 were used for evaluation, and the measurement was repeated 6 times for the respective samples. The obtained results of quantitative analysis as well as the repeatability (CV value) are shown in Table 5.

Query 2: In Fig.4 the calibration curve range 250 to 1500 $\mu\text{g/g}$ (ppm). But in the Table.5 the repeatability results are more than calibration range. As per quantification rule the results are within the calibration range.

→ It is also common to your Query 1, and everything depends on the ineligibility of Fig.4. With the replacement in Fig.4, half of the doubt was solved.

In addition, in this study we use a standard substance of 1000 $\mu\text{g/g}$ and I can understand your perception that it is different from the multipoint calibration curve of the usual calibration curve method. However, this study treated this calibration curve because it maintained two of (1) verification with a minimum process to respond to rapid analysis, (2) conforming to international standards method.

Supplementary explanation for (2) above is as follows. As a regulation method of PhEs screening analysis by Py/TD-GC-MS described in International Standard of IEC 62321-8, Blank sample is used for "Check contamination and carry over" verification, 100 mg/kg standard substance for " check the sensitivity ", and it stipulates that 1000 mg / kg standard substance is used for " Calibration ". And in this regulation, standardized to screen analysis of PhEs at 500 - 2000 mg/kg using standard substances of this concentration. In this research, we carried out the process according to this method. We accepted your proposal and judged that above explanation is necessary in the text, then added it in the text of 4-2.

Query 3: In Fig.2 shows the Thermal desorption peak profile of SIM mode measurement of DBP, BBP & DEHP. But there is no Thermal desorption peak of DBDE observed.

Query 4: In Fig.4 the calibration curves of DBP, BBP & DEHP prepared using powder reference material. But there is no calibration a curve of DBDE was observed even though the reference material was available. (As you mentioned in manuscript).

→ We answer both Query 3 and Query 4 together.

Fig.2 shows the measurement results of DBP, BBP and DEHP contained samples, which are not detected because DBDE do not contain.

In this research, we conducted the following evaluations in order using our new apparatus. (1) Investigation of the detectability of DBP, BBP, and DEHP (Detectable, how short the measurement time, usability of the calibration curve). (2) respective DBP + DIBP, BBP, and DEHP + DNOP were prepared in PE matrix, and the effectiveness of the apparatus for analysis of these samples was evaluated, (3) We prepare samples of respective DBP, BBP, and DEHP containing 500, 1000, 1500 and 2000 µg/g in PVC and investigated the correlation between analysis results and the prepared concentration values, (4) Investigation of the extensibility of our analysis subjects (In the present study, since bromodiphenyl ethers is also subject to regulation in RoHS2, confirm whether this substance can be detected).

Therefore, for DBDE related to (4), theoretical proposal was made to evaluate whether there is detection capability (qualitative analysis) and quantitative discrimination, and quantitative evaluation by the calibration curve method, such as shown in Fig.4 using different concentrations of DBDEs, was not carried out.

However, here we show that several spectral DBDE patterns can be obtained within 15 minutes, and furthermore we found that it has a high correlation between the theoretical spectral patterns and the measurement spectral patterns, as a result of the theoretical quantitative discrimination by measuring the standard substance of DBDE containing polystyrene.

The reason of our using standard material of polystyrene containing DBDE was important to use a target sample clearly known to contain a certain amount of DBDE.

However, here we show that several spectral DBDE patterns can be obtained within 15 minutes, and furthermore we found that it has a high correlation between the theoretical spectral patterns and the measured spectral patterns, which theoretical patterns were obtained as a result of the theoretical quantitative discrimination by measuring the standard substance of DBDE containing polystyrene.

According your suggestion, we described the process (1) - (4) above mentioned as an outline at the beginning of "Materials and Methods", in order to avoid confusion between the various experimental processes mentioned above and the description of the result.